# Few-Shot User-Adaptable Radar-Based Breath Signal Sensing

**DOI:** 10.3390/s23020804

**Published:** 2023-01-10

**Authors:** Gianfranco Mauro, Maria De Carlos Diez, Julius Ott, Lorenzo Servadei, Manuel P. Cuellar, Diego P. Morales-Santos

**Affiliations:** 1Infineon Technologies AG, Am Campeon 1-15, 85579 Neubiberg, Germany; 2Department of Electronic and Computer Technology, University of Granada, Avenida de Fuente Nueva s/n, 18071 Granada, Spain; 3Department of Electrical and Computer Engineering, Technical University of Munich, Arcisstrasse 21, 80333 Munich, Germany; 4Department of Computer Science and Artificial Intelligence, University of Granada, C/. Pdta. Daniel Saucedo Aranda s/n, 18015 Granada, Spain

**Keywords:** vital sign sensing, respiration signal, artificial neural networks, meta-learning, radar, FMCW, few-shot learning, autocorrelation, variational autoencoder, signal processing

## Abstract

Vital signs estimation provides valuable information about an individual’s overall health status. Gathering such information usually requires wearable devices or privacy-invasive settings. In this work, we propose a radar-based user-adaptable solution for respiratory signal prediction while sitting at an office desk. Such an approach leads to a contact-free, privacy-friendly, and easily adaptable system with little reference training data. Data from 24 subjects are preprocessed to extract respiration information using a 60 GHz frequency-modulated continuous wave radar. With few training examples, episodic optimization-based learning allows for generalization to new individuals. Episodically, a convolutional variational autoencoder learns how to map the processed radar data to a reference signal, generating a constrained latent space to the central respiration frequency. Moreover, autocorrelation over recorded radar data time assesses the information corruption due to subject motions. The model learning procedure and breathing prediction are adjusted by exploiting the motion corruption level. Thanks to the episodic acquired knowledge, the model requires an adaptation time of less than one and two seconds for one to five training examples, respectively. The suggested approach represents a novel, quickly adaptable, non-contact alternative for office settings with little user motion.

## 1. Introduction

Estimating a person’s vital parameters has always been an important research topic, as it allows tracking of health status and preventing some diseases and potential accidents [1,2]. Vital signs include the breath wave, heartbeat, body temperature, and blood pressure. The main focus of research is the heart wave, which gives direct information about how a person’s heart is working and can help prevent life-threatening events such as a heart attack or arrhythmia. The breath signal can instead provide information on how the lungs are behaving. The breath wave shape can highlight if the subject is undergoing a hyperventilation episode or if the airways are obstructed due to an allergic reaction or a physical blockage [3,4]. The estimation of vital parameters is usually performed to diagnose a health problem caused by some often acute symptoms. On the other hand, continuous vital sign monitoring could predict and prevent the worsening of respiratory and cardiovascular diseases, which account for 32% of worldwide deaths per year [5]. Vital parameter estimation can be performed over days with portable ambulatory devices such as the Holter monitor for electrocardiogram (ECG). For long-term measurements intended for prevention, however, ambulatory machines are not versatile due to cost, maintenance, and the limitations of user activities. To counter this, wearable devices capable of monitoring multiple vital parameters at the same time and reporting abnormalities have emerged over the years [6,7]. Many wearable devices, such as smartwatches, proved to help predict vital anomalies, but they also have the intrinsic need to be continuously worn. This can be hard in the case, for example, of bulkier devices for breathing sense, which can be worn by newborns or elders. Many solutions are therefore moving toward non-contact sensing techniques [8].

Some non-contact solutions employ camera sensors. Through video signal processing, it is indeed possible to extract parameters such as heart rate (HR) and respiration rate (RR), which can be particularly useful in clinical or telehealth consultations [9,10]. The use of camera sensors, however, can be inadequate in many applications, leading, especially in long-term monitoring, to serious privacy concerns. The use of thermal sensors can be employed to partially overcome this problem [11,12]. Yet, thermal measurements are sensitive to heat and weather conditions and are not employable in all contexts. For contact-free sensing of vital parameters, ultrasound systems can be an excellent privacy-friendly solution [13]. High-frequency systems such as radar or Wi-Fi may have additional advantages, such as a much greater spatial range and the ability to pass through surfaces [14,15,16]. WiFi-based solutions can be very accurate in estimating vital signs [17,18] but often require systems with transmitting (Tx) and receiving (Rx) antennas placed in separate devices, contributing to higher power consumption than radar. Remarkable among the various radar modulations is the frequency modulated continuous wave (FMCW), which enables simultaneous estimation of the relative range, velocity, and angle of arrival of targets placed in the sensor’s field of view (FoV) [19,20]. The ability to sense static components, thanks to the frequency modulation of chirp signals sent from the Tx channels of the FCMW radar, can enable privacy-friendly tracking of targets in the FoV. Further, thanks to the micro-Doppler effect, radar can also sense small and periodic displacements generated as vital signs [21]. The collected information, preprocessed in phase, is particularly corruptible by continuous user movement. Nevertheless, non-contact vital parameter estimation can be employed in relatively static settings, such as an office, even for multi-person sensing [22].

Raw radar data are inherently difficult to interpret and often require artificial intelligence (AI) techniques to filter useful information rather than pure signal processing or computer vision. Many state-of-the-art solutions use Kalman filters to reduce measured noise in vital signs or to update the specific parameter band-pass filter limits for estimates and uncertainties of chosen state variables [23,24,25]. However, given the Kalman filter assumptions, it is necessary to selectively filter out corrupted data caused by random user movements to avoid corrupting subsequent vital sign estimates [23]. Other solutions employ machine learning (ML) approaches to predict vital parameters in the form of time series [26] or to extract relevant information, such as arrhythmia detection [27]. In other cases, the interest is more in estimating the number of peaks in time than in reconstructing the vital signs. In [28], to decrease prediction latency, the solution uses an artificial neural network (ANN) exclusively to predict the presence of heart peaks from raw radar data using labeled ECGs. Although all AI-listed solutions enable accurate reconstructions of vital parameters or estimation of target variables, they all require a large dataset of training data on various subjects for such an achievement. While tested in various contexts and for new users, many solutions may not easily fit users with unconventional vital signs or in new recording positions and angles. To address the big data need, a specific branch of ML called Meta-Learning (Meta-L) is gaining momentum [29]. In Meta-L, the goal is context generalization. An algorithm learns to solve various tasks via episodes, leveraging the accumulated experience and only a little new available data to adapt faster in new contexts. Usually, in each episode, the model tries to address an *N–ways* task, where *N* represents the number of classes (one per regression) and *k–shots*, or *k* examples per class. Generally, tasks are sampled from the same domain, and episodic learning often occurs in two phases. In an inner step, the learning algorithm learns to solve a given task by exploiting support (*S*) examples. In an outer step, the ability to generalize to new tasks is estimated and tuned using query (*Q*) examples. In optimization-based algorithms, such as model-agnostic meta-learning (MAML) [30], the learning algorithm parameters update is performed using gradient descent in both episodic steps.

In this paper, we present a user-adaptable FMCW radar solution based on signal processing and Meta-L for breath signal estimation. The approach was tested in an office desk-workplace with a single person in the FoV. The ideal user-radar board distance is up to 40 cm. This non-contact solution is employable when the user under test performs some actions characterized by little movements, such as laughing, talking, or using the keyboard, but leads to better performances in idle scenarios. Continuous detection of the user-radar range enables user tracking and radar preprocessing adaptation. Through prior information acquired via Meta-L, the algorithm is adaptable to a new person with a single or few training examples. The episodic training is designed to extract the breath information from the radar data while minimizing the contribution of the detected motion corruption due to the user’s actions. Data are collected for short sessions at the desk-workplace, from 24 different users at 2 ranges of distances, up to 30 cm and 40 cm. Among all the users, 14 are selected for training and 10 for testing. Radar data are gathered via the FMCW 60 GHz radar system with 1 Tx and 3 Rx, while a breath-sensing belt is used as a reference. In a single 30-second session, the collected radar data are first preprocessed in frequency to extract the range information of a single target user. The phase signal, which contains the breath information, is then unwrapped for a selected set of range bins, which are dynamically adjusted over the sessions. A multi-output ANN trained episodically in 1–, 5–, and 10–shots aims to predict the user’s breath signal from the phase signal. The ANN maps, via a convolutional variational autoencoder (C-VAE), the radar phase to the belt reference signal, constraining the generation of latent space to the central frequency (Fc) of breath. The overall topology scheme is depicted in Figure 1. A series of two-band-pass digital biquad filters selectively filter the breath signal information according to the predicted Fc. The autocorrelation of the extracted signal, performed with a sliding window, allows estimating per session the level of corruption due to user motion. The corruption information is thus employed in both episodic training and prediction to improve ANN performance by fetching the most valuable information available in motion-corrupted sessions. The main contributions of this paper are as follows:1.Implementation, to the best of our knowledge, of the first few-shot user-adaptable radar-based breath signal sensing solution.2.Development of a specialized radar data preprocessing pipeline that dynamically tracks the user’s position relative to the board.3.Design of a cost function that constrains the generation of the latent space of a C-VAE to the respiration Fc in a multi-output ANN.4.Development of a corruption-based sample weighting approach that guides the breathing signal estimation in the presence of user motion.

## 2. Related Works

In this section, we first investigate methods for non-contact estimation of the breath signal, focusing mainly on high-frequency solutions that are AI-oriented. We then discuss vital sign-sensing approaches that employ Meta-L techniques.

Alizadeh et al. [31] used a 77 GHz FMCW radar to extract vital parameters from a patient lying down on a bed in a non-contact vital sign sensing solution. In this work, vital signs are estimated by purely signal-processing-based methods. An initial fast Fourier transform (FFT) is performed to extract the range information of the subject from the radar board. The Fc of the vital signs is estimated on the unwrapped phase signal downstream of a second FFT that calculates the vibrations, leading to the generation of a range-vibration map. The vital signs are then extracted via a band-pass filter. This method allows reconstruction of breath rate 94% similar to a reference signal but places the major constraint that the only non-stationary features in the range-vibration map are the biological activities. This makes the solution only applicable when the subject under test is idle. Wang Y. et al. [32] proposed two different methods of vital signal estimation from phase information extracted from data collected with a 77 GHz FMCW. These methods, namely the Compressive Sensing based on orthogonal matching pursuit (CS-OMP) algorithm and the Rigrsure Adaptive soft threshold noise reduction based on discrete wavelet transform (RA-DWT), separate and reconstruct breathing and heartbeat signals instead of the more traditional band-pass filtering. Although the results obtained are very similar to those obtained with contact-based reference sensors, there is still the inherent constraint that the subject has to remain stationary in front of the radar system. Iyer et al. [27] developed a solution that uses Fourier series analysis on data collected by a 77 GHz FMCW radar to extract the vital signs of an individual from various orientations. Although the paper mainly focuses on the heartbeat for detecting arrhythmias using an ANN, the breath rate (BR) and the breathing wave are also estimated. The latter is obtained through a digital biquad band-pass filter whose parameters are invariant to the user or recording session. A filter that is not selective enough can lead to noisy predictions with many falsely detected breath peaks due to motion. Lee et al. [33] implemented a solution that detects the vital parameters of multiple subjects in the FoV using a 24 GHz FMCW Doppler radar. Doppler phase information is combined with range measurements obtained by parametric spectral estimation to distinguish multiple targets even beyond the theoretical range resolution limit. Likewise, in this approach, a band-pass filter with relatively wide bandwidth is utilized, which may not be adequate in all contexts. Lv et al. [34] used a much higher frequency FMCW 120 GHz radar system to estimate the vital signs of eight volunteers. The solution mainly focuses on acquiring the heartbeat signal, utilizing a notch filter to filter out the respiratory harmonics in the spectrum of interest. This is mainly conducted to overcome the problem of overlapping and interference of breathing and heart harmonics in some measurements. As also mentioned by the authors, the classical FFT approach does not guarantee the correct prediction of vital parameters in motion-corrupted scenarios. Gong et al. [35] illustrate an FMCW-based solution for vital sign estimation that also seeks to address the problem of sensing even in the presence of motion. The approach combines direct FMCW sensing for static instances with an indirect vital sign prediction based on motion power estimation. Two sub-long short-term memories (sub-LSTMs) are used to estimate the RR; they first classify the motion patterns and then estimate the RR. The method is robust even with some random movement patterns, such as lifting an arm, in new environments, and with new users. However, the variation in RR is estimated and not the respiratory signal, which can give additional information about a user’s health quality. It is also not specified whether the users were allowed to speak during the recordings or whether this activity was taken into account. Wang D. et al. [36] proposed an interesting comparison in vital sign estimation between impulse radio ultra-wideband (IR-UWB) and FMCW radar. While radar FMCW needs phase information to extract vital signs, IR-UWB uses distance information. The data of both radar topologies are processed with a relatively standard approach that employs a band-pass filter. The IR-UWB achieves a better estimate and signal-to-noise ratio (SNR) but needs to send many pulses to distinguish the signal from noise. A high pulse rate per second also requires a high-speed analog-to-digital converter (ADC) which increases cost and hardware design complexity compared to the FMCW. For the FMCW, on the other hand, a narrow instantaneous bandwidth allows the use of lower-speed ADCs. In addition, multiple-input-multiple-output (MIMO) topologies for FMCWs allow multiple target locations and real-time monitoring. Rana et al. [37] presented a system that processes, via short-term Fourier transformation (STFT), the data collected from a UWB radar to extract vital signs. Data are collected in various areas of the house. The UWB recordings are complemented by a multi-class support vector machine (MC-SVM) that distinguishes vital signs when different activities are performed in the available locations. This approach shows preliminary results of how it is possible to recognize specific user activities with little training data. This could also potentially be used to improve the estimation of activity-related vital signs. Khan et al. [38] illustrated a channel state information (CSI) based WiFi sensing solution to track the vital signs of a patient. With the features extracted from the collected data, the health status of patients is estimated through four types of ML algorithms via classification. These algorithms are K-nearest neighbor (KNN), decision tree, random forest, and support vector machine (SVM). The presented feature extraction approaches make it possible to preserve valid information by decreasing the dimension of the individual examples collected and simplifying the task of ML algorithms.

As far as we know, there is only one source for video-based physiological measurement that uses Meta-L and few-shot learning to estimate vital parameters. Liu et al. [39] proposed a Meta-L-based approach for personalized video-based non-contact cardiac pulse and heart rate monitoring. Thanks to the episodic training of MAML, the approach requires only 18 seconds of video for customization to new scenarios with different users, sunlight, and indoor illuminations. The solution, evaluated in two benchmark datasets, yielded substantially superior performances compared to state-of-the-art approaches.

## 3. System Description and Implementation

This section gives a general overview of the system, a description of how the data acquisition system is set up, details on radar system configuration, and the main preprocessing steps.

### 3.1. General Overview of the Proposed Framework

The proposed framework is depicted in Figure 2. For a Meta-L solution, a dataset as small and diverse as possible is generated. Specific information about the breath dataset for Meta-L is given in Section 3.7. The recordings from the respiration belt, which serves as a reference sensor, and the FMCW radar are collected synchronously over 30 s sessions using the recording setup described in Section 3.3. The preprocessing, as depicted in Section 3.4, aims to unwrap the phase information, reconstructing the displacement generated by user breathing. During dataset generation, the radar-based respiration signal Fc is estimated by calculating the maximum correlation between the radar phase and belt reference downstream of a double biquad band-pass filter (Section 3.5.1). The entire Meta-L stage is presented in Section 4. From the estimated breath signal, it is also possible to calculate the instantaneous breaths per minute (bpm) and the amount of corruption per recording session caused by user motion (Section 3.6).

### 3.2. Radar Board and Configuration

The chosen radar system for this application is the XENSIV™ 60 GHz *BGT60TR13* FMCW, manufactured by Infineon Technologies AG [40]. The radar board is a miniaturized and low-power frequency modulated solution with a center frequency $f_{0}$ of 60 GHz and a bandwidth of approximately 6 GHz, which allows for a high range resolution of approximately 2 cm. In sensing applications within 5 m, the power consumption is reduced to only 5 mW thanks to an operation-optimized duty cycle. Further, by exploiting the micro-Doppler effect through phase analysis, it is possible to capture periodic displacements over time, such as vital signs, well below the 2-cm range limit [21]. The *BGT60TR13C* has three Rx channels and one Tx channel, all embedded in the package. Additionally, to enable accurate estimation of targets’ azimuth and elevation angles of arrival (AoAs) in the FoV, the Rx antennas are positioned orthogonally to each other. With an $f_{0}$ of 60 GHz and a single Tx channel, such a board provides a less expensive and lower-frequency solution than many cutting-edge non-contact high-frequency vital signs systems. The evaluation board with the sensor board mounted on top is shown in Figure 3.

The *BGT60TR13C* generates chirps, which are a series of linearly frequency-modulated signals with a bandwidth of $B_{w}$ centered on $f_{0}$. Each chirp lasts $t_{c}$ and is made up of a predetermined number of $n_{s}$ samples. In use, the data gathered from the Rx channels are mixed with a Tx reference and digitized with 12-bit resolution. The generated output signal is referred to as intermediate frequency (IF). Radar data are frequently compressed into frames for additional preprocessing, with each frame carrying the IF for a series of $N_{c}$ chirps. For an FMCW modulation, the theoretical range resolution Δr and maximum detection range $R_{max}$ are calculated using the following formulas:(1)$$\Delta r=\frac{c}{2B_{w}}\;,$$
(2)$$R_{max}=\frac{\Delta r}{2}n_{s}\;,$$
where *c* indicates the speed of light in the air. For the application of breath sensing at the workplace desk, a theoretical $R_{max}$ of 50 cm would be sufficient. However, a theoretical maximum distance of about 3.75 m was selected for compatibility with other use cases and for future works. In the preprocessing, though, only the range bins where the user is detected are processed. The selected Δr instead is roughly 37.5 cm, which enables user identification from the surrounding clutter (static targets). The set values of $B_{w}$ and $n_{s}$ are accordingly 4 GHz and 200. For appropriate phase analysis, we also chose $t_{c}$ and $N_{c}$ values of 150 μs and 2, respectively. To acquire around 20 frames per second, a frame repetition time (fps) of 50 ms was chosen. Additionally, a 2 MHz ADC sampling rate $F_{s}$ was used. All the values selected for the radar board configuration are outlined in Table 1.

### 3.3. Recording Setup

The recording setup, shown in Figure 4, is consistent with the chosen application, i.e., at-desk workplace monitoring at short distances (up to about 40 cm). The *BGT60TR13C* radar system is mounted on the front of the desk, and the Go Direct ® respiration belt [41] is placed at the level of the users’ diaphragm. The belt is used as a reference to measure displacement in N (Newtons). We chose to use this belt as a reference since it is employed in other state-of-the-art work for benchmarking with radar solutions such as [35,36]. As reported in these works, the belt has a force resolution of 0.1 Newton. This resolution allows displacements generated by breathing to be distinguishable in the presence of user motion. Although we consider the belt as a reference, such a sensor may also be subject to motion corruption. In the specific use case at the desk workplace, many of the movements made by users have little impact on a chest-mounted wearable sensor. A practical example may be typing on the keyboard. Other movements, such as bending the back, can also degrade the belt signal. In our work, however, we impose the constraint that the belt signal is the ground truth, unaffected by motion corruption noise. The data gathered by the respiration belt and radar system have been synchronized during the recordings. The data from the two sensors were synchronized frame by frame using a global time stamp generated at the laptop level. The gathering and synchronization have been performed with an Intel^®^ Core i7-8700K CPU. Data collection was performed for 24 healthy users with an age range of up to about 35 years. All users agreed in advance to participate in data collection. The data were collected and stored as anonymously as possible, without tracking names or other characteristics that could be used to identify an individual. The data will not be made public. The users were told to behave as normally as possible, performing actions such as laughing, joking, and using the keyboard and mouse. Many users also chose to watch a video during data collection to avoid respiratory bias due to recording. For each user, 20 sessions of 30 s each were collected. Two desks in different offices and two distance ranges were chosen. The desks used are of the same type and height (about 76 cm). However, data were collected in two different environments to avoid the potential overfitting of ML models on a single location. A total of 10 sessions per user were collected at a distance from the radar board to the person’s chest of up to 30 cm and another 10 up to about 40 cm. With the 10 min allotted to each user, a total of 4 h of data was collected.

### 3.4. Radar Phase Signal Extraction

Thanks to the micro-Doppler effect [21], it is possible to extract the breath information from the unwrapped phase signal derived from the raw radar data. The preprocessing pipeline for the application is shown in Figure 5. The preprocessing can be divided into the following steps:Raw radar and respiration belt data are collected synchronously for a session. The chosen frame rate per session is 660 ($N_{m}$), which is 10% higher than the theoretical frame rate of 600 (20 fps * 30 s). Longer sessions for either sensor are interpolated, whereas shorter ones are zero-padded. The belt signal is used as a reference estimation in the Meta-L training phase. Subsequent preprocessing steps involve the radar signal only.The IF signal is computed channel-wise, for the three Rx, for each radar frame. The information is organized in a 3D matrix, with the x-axis representing fast time (samples), the y-axis representing slow time (chirps), and the z-axis representing channels.The average value is subtracted from the sequence of 660 frames so that the potential direct current (DC) offset is subtracted.Over slow time and channels, the radar-sensed information derives from the same recorded event. Rather than using a single channel or single chirp, we use the averaged information over both axes for the next steps. Intrinsically, given the equal importance of the information in the chirps and their respective channels, the averaged information will be more robust to the noise.A 1D FFT is performed along fast-time to retrieve the range information.From the range information, it is possible to estimate the user’s position frame-wise, select the set of meaningful range bins, and subtract the clutter in each (Section 3.5).The phase information is calculated for the selected bins. Frame-wise, only the bin range with the highest mean squared error (MSE) to the estimated clutter is chosen (Section 3.5).The phase beyond (−pi,pi) is then unwrapped using a phase discontinuity threshold approach.Because users had freedom of action during the recordings, the Fc estimated from the radar phase by frequency analysis may not coincide with the central respiration Fc. For this reason, Meta-L is used to map the radar phase to the computed ideal belt Fc (Section 3.5.1).Comparison between radar-estimated breath signal and respiration belt is performed on normalized signals between zero and one, calculating MSE and estimating instantaneous bpm along the session (Section 3.6).

### 3.5. Range Bins Selection and Clutter Removal

Relevant radar information is only contained in a limited range of bins that reflect the user’s position relative to the radar board. Let $S_{R}\left(m,s\right)$ with $m\in [0,N_{m}]$ and $s\in [0,n_{s}]$ be the radar signal with range information on the x-axis and slow time on the y-axis. The maximum bin range is calculated ∀m as follows:(3)$$\max_{s}\left|S_{R}\left(m,s\right)\right|\;.$$

Around the maximum detected, 12 range bins are also processed for phase information extraction. The boundary range bins are dynamically updated via a moving average of eight frames. The dynamic adaptation avoids abrupt changes in the range under process due to instantaneous noise. Clutter is computed only for the selected bins *s*, frame-wise $\forall \;m\in [0,N_{m}]$, using the moving target indication (MTI):(4)$${\bar{S_{R}\left(s\right)}}_{new}=\alpha \;S_{R}\left(m,s\right)+\left(1-\alpha \right)\;{\bar{S_{R}\left(s\right)}}_{old}\;;$$
where α∈[0,1] is set to 0.4 and ${\bar{S_{R}\left(s\right)}}_{old}$ is the average over the preceding *s* bins. The value of α was chosen empirically, noting that values less than 0.2 gave too much weight to previous clutter contributions, while values greater than 0.6 depended too much on the current radar signal. Only the bin range with the highest peak-to-clutter information is extracted, which corresponds theoretically to the subject position at a given time. The MSE between $S_{R}\left(m,s\right)$ and the new clutter ${\bar{S_{R}\left(s\right)}}_{new}$ is calculated. The maximum MSE value corresponds to the highest peak-to-clutter. Two examples of the user range over session time are depicted in Figure 6.

#### 3.5.1. Central Frequency Estimation and Labeling

The respiration rate can be estimated by spectral analysis in a given session, for example, by analyzing the power spectrum and taking the maximum peak in a given frequency range. Such a method is often employed for radar data recorded in idle conditions but is more of a challenge in the presence of user motion. The respiration bandwidth and Fc also depend strongly on the physiology and characteristics of the individual. Therefore, we propose using Meta-L to estimate the Fc in a user-adaptable manner, using the central frequency extracted from the respiration belt as a reference label. The reference Fc is obtained from the belt signal power spectrum by locating the frequency corresponding to the maximum peak in the limit range [0.1, 0.5] Hz, corresponding to 6 and 30 bpm. Section 4 describes how the proposed method estimates radar-based breathing signal learning from both the reference belt signal and relative Fc. The radar-based breathing signal is then computed by applying a sequence of two biquad band-pass filters to the unwrapped phase signal with the estimated Fc. The employed filter is a second-order digital recursive linear infinite impulse response (IIR) containing two poles and two zeros. A time representation of the filter can be described as follows:(5)$$O\left[n\right]=a_{0}\;I\left[n\right]+a_{1}\;I\left[n-1\right]+a_{2}\;I\left[n-2\right]-b_{1}\;O\left[n-1\right]-b_{2}\;O\left[n-2\right]\;,$$
where *n* is the time step; *I* the input vector; *O* the output vector, and $a_{0},\;a_{1},\;a_{2},\;b_{0},\;b_{1},\;b_{2}$, the filter parameters according to the type. These latter parameters depend on the Fc selected for a given session. The formulas are provided in Appendix A. Each of the two cascaded biquad filters has a quality factor *Q* of $\sqrt{2}$ and a sampling frequency (fs) of 20 Hz, corresponding to the fps. The characteristics of the filter are outlined in Figure 7.

### 3.6. Breaths per Minute Estimation and Corruption Detection

Along with the collected data session, the instantaneous bpm can be assessed via a sliding window. This information may also be useful in radar sessions that have been partially corrupted by motion and contain less respiration information. The sliding window is dynamically computed per session proportionally to the average distance between peaks. By leveraging this window, it is also possible to estimate the instantaneous motion corruption by comparing the signal with itself through autocorrelation. The corruption information is used to weight the training samples in Meta-L and improve the predictions, as explained in Section 4.

In a recording session, the sliding window is defined as twice the average distance between the detected peaks of the radar phase signal after band-pass filtering. The window length is, therefore, about two whole cycles of breathing, intended as sequences of inhalation and exhalation. Because the estimated peaks for radar and belt may not match, a specific sliding window is calculated for each of the two signals. An example of the belt and radar respiration signal with computed sliding window is shown in Figure 8. In this instance, the respiration Fc for the radar signal is ideally extracted from the belt and has not yet been estimated by Meta-L. Local peak time shifts are visible in the plot between the radar and belt signals. These shifts are caused by two main reasons in the radar signal. First, the belt signal already contains the breathing information, whereas the radar requires multiple preprocessing steps. These steps, including the biquad filter, cause global shifts in the extracted information. In addition, the respiration belt is connected to the individual during recordings, while the radar is connected to the desk. As a result, millimeter-scale user displacements along the session can contribute to local shifts in the radar respiration signal peaks with respect to the belt. Discrepancies in amplitude, on the other hand, can be caused by ambient noise and the extracted phase, which is very sensitive to small displacements. Corruption is also visible in the radar signal at the beginning of the session. As it is not present in the belt signal, it was most likely caused by arm movements, which are mostly undetectable by the wearable sensor on the chest.

The instantaneous bpm value is estimated as the number of peaks within the sliding window throughout the session. Because two different sliding windows are calculated for radar and belt, the length of the x-axis bpm estimate (number of samples minus the length of the sliding window) may not match in all the sessions. Autocorrelation along the session is used to estimate corruption, with a window about one and a half times as long as a breathing peak (three-quarters of a bpm sliding window). The correlation of the signal with itself gives a measure of how similar and periodic it is over time. A flag variable, by default set to zero, is set to one when the maximum autocorrelation goes below a threshold. This threshold is adjusted dynamically for the length of the sliding window. Empirically, this value is, for normalized sessions between 0 and 1, set to 0.001 times the sliding window length. On user-collected test examples, a magnitude less than 0.001 or greater seems to lead to over- or underestimation of corruption, respectively. The instantaneous bpm and corruption flag are plotted in Figure 9 for the same session as in Figure 8.

### 3.7. Breath Meta-Dataset

The *Breath Meta-Dataset* for training and testing the Meta-L algorithm contains data collected from 24 different healthy users up to 35 years old. Specific information on setup and data collection is provided in Section 3.3. For each session collected, the unwrapped radar phase signal (Meta-L input), the respiration belt signal, and the corresponding ideal respiration Fc (Meta-L outputs) are saved. All signals are interpolated or sampled to have a length of 660 samples for the 30-second recording. Both radar phase and belt signals are normalized between zero and one and translated in the interval by their average. Fourteen users were randomly selected for episodic Meta-L training, whereas the other ten were used for testing. The breath signal is highly dependent on an individual’s characteristics and the presence of motion. A subject-wise two-component t-distributed stochastic neighbor embedding (t-SNE [42]) of the radar phase signal for all data collected is shown in Figure 10. All radar signals for t-SNE representation were filtered with a series of 2 band-pass filters with respiration frequency Fc fixed at 0.33 Hz and Q at 0.8. As can be seen from the figure, under t-SNE assumptions, two components do not seem to be sufficient to show user-specific characteristics. This emphasizes how complex the interpretation of radar data is to extract features in an unsupervised manner for such an application.

## 4. Proposed Method

In this section, we describe the algorithm and topology we chose to generate the user-adaptable Meta-L model on the *Breath Meta-Dataset*. We propose an episodic learning approach by exploiting Meta-L and a C-VAE for regularized feature extraction. Once trained for generalization, leveraging a few examples gathered via the reference respiration belt, the model enables the fast adaptation of the non-contact radar sensing solution to a new user. To partially overcome the problem of motion corruption in sessions, we also present an optimized loss function (Section 4.3) that makes use of the corruption estimation method presented in Section 3.6.

### 4.1. Episodic Breath Signal Estimation

For the episodic breath signal estimation approach, we use the optimization-based MAML second-order algorithm (MAML $\;2^{nd}$) [30]. Let *R* be the set of training episodes. A task $T_{r}$ is sampled for each r∈R, corresponding to a single training user. During the episode, a model learns to map the unwrapped radar phase *x* to the reference data of the respiration belt $x_{belt}$ using *k* shots of support. The model learns by minimizing the binary cross-entropy (BCE), as follows:(6)$$BCE\left(x,x_{belt}\right)=-x\;log\left(x_{belt}\right)-\left(1-x\right)\;log\left(1-x_{belt}\right),$$
with variables *x* and $x_{belt}$ in [0,1].

Radar features are encoded in a normally distributed variable $z∼N(\mu_{x},\sigma_{x})$ using a C-VAE topology. The $\mu_{x}$ and $\sigma_{x}$ variables represent the mean and standard deviation for a given latent space dimension and input *x*, respectively. In addition, the Kullback–Leibler (KL) divergence [43] is used to ensure that *z* is close to a reference N(0,1) distribution. Although KL divergence allows regularization of latent space approaching a standard multivariate normal distribution, C-VAE could learn to extract unnecessary information from the unwrapped radar phase. Such information includes, for example, displacements caused by user motion during sessions or noise. To overcome this, the latent space generation is constrained by breathing information. This is achieved by minimizing the MSE between ideal Fc extracted from the belt (*y*) and $\hat{y}$ predicted via a single-neuron dense layer with the linear activation function.

Adding up the three components, the loss function *L* is defined as follows:(7)$$L\left(x,\;x_{belt},\;y,\;\hat{y}\right)=BCE\left(x,\;x_{belt}\right)+KL\left[N\left(\mu_{x},\;\sigma_{x}\right),\;N\left(0,\;1\right)\right]+K\;||y-\hat{y}{\left|\right|}^{2},$$
where *K* equal to 1000 is an equalization coefficient aimed at adjusting the magnitude of the MSE. The same loss function is also used in the outer step of Meta-L, on a query sample, given for the same task $T_{r}$. The loss function terms are represented in Figure 11.

For a fixed training strategy, model generalization is assessed based on the ability to perform better on new tasks as episodes progress. This is performed by evaluating the model after each outer step on two evaluation tasks $T_{r}$ and one $T_{v}$ sampled by the training and test users, respectively. Box plots are constructed based on the loss values obtained for sequences of episodes. As the episodes progress, the mean loss should decrease, and the box plots’ interquartile range (IQR) and whiskers should also get smaller. This represents the ideal training behavior in Meta-L. Such factors, when also observed on the tasks $T_{v}$, highlight how generalization occurs even on test users, never observed in the training phase.

### 4.2. Proposed C-VAE-Based Topology

The chosen C-VAE topology takes the unwrapped radar phase *x* as an input and returns two outputs. Decoder-side, the network attempts to reconstruct the $x_{belt}$ reference signal from the variable *z*. The ideal Fc, also extracted from the belt reference, is regressed from the latent space, with a single-neuron dense layer and linear activation function. The encoder contains two sequences of convolutional blocks that extract features from *x*. The decoder, on the other hand, tries to reconstruct $x_{belt}$ starting from *z* with two deconvolution blocks and up-sampling. The C-VAE topology is shown in Figure 12 with an indication of layers and their parameters. The strategy of mapping the unwrapped radar phase to the belt signal attempts to counteract the problems of amplitude discrepancy and local peak time shift described in Section 3.6. The latent space generated during training, in fact, depends on the belt signal, which is gathered directly from the sensor attached to the individual’s lower chest.

The dimension of the latent space can considerably impact the model’s performance. In our experiments, we chose a dimension of 32 as the trade-off between performance and topology size. In total, the chosen topology has 739,074 parameters, all of which are trainable. Some examples of generated latent space in relation to different inputs are shown in Figure 13.

### 4.3. Corruption-Weighted Loss and Breathing Estimation Formulation

Even though C-VAE is set up to get information about breathing by predicting the Fc, there is still a problem when the user is moving. Radar data sessions may, in fact, be highly corrupted by motion noise and not contain the necessary respiration information. In such cases, mapping the unwrapped radar phase to the belt signal is not an effective choice. With the amount of predicted corruption motion and taking advantage of the method described in Section 3.6, a higher priority can be given to estimating the Fc than to reconstructing the ideal signal. The corruption rate for each session can be estimated by summing frame-wise $\forall \;m\in [0,N_{m}]$ the corruption flag variable c(m). The greater the motion corruption, the greater the contribution of Fc in the Loss Function *L* must be over the signal reconstruction term.

The *L* can then be adjusted as follows ($L^{*})$: (8)$$L^{*}=\tau \;BCE\left(x,\;x_{belt}\right)+KL\left[N\left(\mu_{x},\;\sigma_{x}\right),N\left(0,\;1\right)\right]+\gamma \;K\;||y-\hat{y}{\left|\right|}^{2},$$
where $\tau =\frac{1}{\sum {}_{m}^{N_{m}}c\left(m\right)}$, and γ=1−τ.

The τ values are obtained during Meta-L training and are normalized per epoch to the training batch size. Consistently, the reconstructed breathing signal via the C-VAE topology can be corrected using the two estimated outputs ${\hat{x}}_{belt}$ and $\hat{y}$ and the predicted corruption level for the single session. An adjusted estimate ${\hat{x}}^{*}$ of the radar-based breathing signal can be given by the following formula:(9)$${\hat{x}}^{*}=\frac{\tau \;{\hat{x}}_{belt}+\gamma \;\epsilon \;biquad\left(x,\hat{y}\right)}{\tau +\gamma \;\epsilon }\;$$
where $biquad(x,\hat{y})$ represents the filtered version of *x*, with the estimated $\hat{y}$ as Fc (Section 3.5.1) and ϵ set to two, makes the contribution of $\hat{y}$ even more dominant in the presence of motion corruption.

### 4.4. Information about Experiments

All experiments have been conducted by minimizing the loss function $L^{*}$, training the C-VAE model with latent dimension 32. As described in Section 4.1 and Section 4.3, the loss function includes a contribution to the FC as well as a contribution to the reconstruction of the respiration signal. The latter imposes normalized signals in the range [0,1]. The loss consequently has no unit of measurement but can be understood as an absolute value to be minimized with respect to zero. The optimizer chosen is Adam, with $\beta_{1}$ and $\beta_{2}$ equal to 0 and 0.5, respectively. The training is performed on 3000 episodes at 4 epochs per episode. For 1–shot experiments, the batch size is one, and for 5– and 10–shot experiments, it is 5. The inner learning rate is set to 18e−4 for the 1–shot experiments and 8e−4 for the 5– and 10–shot experiments to avoid episodic overfitting. The chosen outer learning rate is 17e−4. Each experiment is performed three times. The performance of the C-VAE model is evaluated in terms of mean $L^{*}$ as episodes progress, adaptation time on new users, and single inference time on a new sample. For the loss evaluation, we also present a confidence value. This value represents the 95 % confidence that the true mean is included in the distribution. In general, the lower this value, the more stable and precise a given type of experiment is. The adaptation time per user corresponds to the time required by the Meta-L model to complete an entire training episode via a simple first-order gradient descent. Optimization is performed for a specific number of epochs and batches by minimizing the loss of $L^{*}$ over *k* training shots. For the adaptation time estimation, we chose four epochs of training with first-order gradient optimization. The other hyperparameters remain unchanged from the Meta-L training procedure. Single-sample inference time is calculated at the end of each adaptation training, on a random sample of tests for a given user. This value is consequently independent of the number of training shots selected in the adaptation. At the end of each adaptation test episode, the model parameters are restored to the values learned during the Meta-L training.

## 5. Results and Discussion

This section presents the results of Meta-L experiments on the *Breath Meta-Dataset*. The experiments were carried out with the optimization-based algorithm (MAML $\;2^{nd}$), for 1–, 5– and 10–shots (Section 5.1). Without, to the best of our knowledge, any state-of-the-art Meta-L solutions for breath sensing, we compare our method to other state-of-the-art Meta-L algorithms, taking advantage of our proposed C-VAE topology (Section 5.3). We then show in an ablative study the benefits of using motion corruption estimation in the loss function and the model performance with various latent dimensions (Section 5.2).

All experiments were performed on Intel^®^ Core i7-8700K CPU, and DIMM 16 GB DDR4-3000 module of RAM.

### 5.1. Results on MAML Second Order

All results presented represent the average of the results achieved in the various repetitions. The model performance was evaluated every 300 episodes, creating a box plot on the collected loss values in the evaluation loop over 10 test examples per class (Section 4.1). The episodic learning trend on a 1–shot experiment is shown in Figure 14. As the episodes progress, $L^{*}$ decreases, as well as the IQR and whiskers. While learning from only one example per user, the model can generalize better thanks to prior acquired experience. This behavior is observable not only for $T_{r}$ tasks but also for $T_{v}$ test tasks, which are unobserved in episodic learning.

Box plots are an effective way to assess the progress of episodic learning but do not reveal the underlying distribution of the $L^{*}$ variable. The histograms built on $L^{*}$ for a given interval of episodes can help estimate the distribution and assess the generalization. The histograms corresponding to the box plots generated for the first and last 300 episodes of a 1–shot experiment are depicted in Figure 15. The bottom plots represent the density histogram of the $L^{*}$, while the Gaussian approximation of the box plots with their respective quartiles is shown in the middle plots. The histograms do not undergo a Gaussian distribution. At the beginning of episodic learning, the distribution is usually multimodal because of the different learning complexity between tasks. In the last learning step, the histograms typically feature a positive skewness toward the zero of the $L^{*}$. This behavior occurs thanks to the generalization of information acquired episodically.

The values of $L^{*}$ obtained for MAML $\;2^{nd}$ experiments for 1–, 5–, and 10–shots on test users are presented in Table 2.

As can be seen from the table, as the number of shots increases, there are no significant reductions in the mean loss for new users. Experiments with 5– and 10–shots, however, show a lower 95 % confidence value, and thus higher precision. This attests that in some cases, a single training example with different characteristics from others gathered may not be sufficient to generalize on the test. The generalization strategy allows the model to extract a substantial amount of breath signal information, independent of the number of shots, as illustrated in Figure 14. Many user data sessions are corrupted by motion and limit the increase in performance of the C-VAE model as the number of training examples increases. However, by relating the loss $L^{*}$ to the average breathing rate per 30 s session, a specific learning behavior can be observed. The box plots generated according to the respiration rate for all test users are shown in Figure 16. The respiration rate between 7 and 9 represents a standard human breathing rate. As can be seen in the figure, the base of the box plots is not uniform over the RR range. In fact, most of the collected examples have a number of respiration peaks that are close to or equivalent to the standard value. Between 1 and 4 and 12 and 14, there are only 4 and 5 examples, respectively. On the other hand, for a respiration rate between 7 and 9, there are about 30 instances per class. This motivates the choice of the box plot construction. By fitting the model to each test user separately after episodic learning and using the remaining sessions as tests for each fit, it is possible to obtain the model behavior as a function of the RR. With the 1–shot fit, the model is more accurate at reconstructing breath signals between 7 and 9 peaks per 30 s. For lower or higher rates, the model performs less well in reconstruction, resulting in an increased loss. This behavior can be due to two main reasons. The first is that motion corruption may not have allowed the identification of the correct breathing peaks in the test sessions. Motion corruption results in the erroneous identification of breathing patterns and subsequent missed learning. The second is that the model did not have enough reference examples for low and high rates during generalization learning. Because of this, a few examples of training for a new user may not be sufficient for adaptation. For the 5– and 10–shots fit, the model seems to be able to tackle low or high RR situations better, but it performs slightly less well than the 1–shot model for standard RRs. This may be mainly caused by motion corruption in many sessions for all users. Although the $L^{*}$ is defined to address such a problem, additional training examples may not contain enough information to overcome motion corruption.

Figure 17 and Figure 18 show examples of prediction after test user adaptation of MAML $\;2^{nd}$ 1–shot. In both figures, the top plots represent the breath signal prediction while the bottom plots represent the instantaneous bpm estimation. The prediction of breath signals is obtained using the ${\hat{x}}^{*}$ formula (Equation (Equation 9)). Figure 17 depicts two examples of correct breath signal prediction throughout the session. In the example (a), there is almost no corruption due to user motion for most of the session. In accordance with Figure 16 for 1–shot, this example falls within the range of 12—14 beats for 30 s. Even so, the algorithm leads to a quite accurate bpm estimation, with an average gap between the belt reference and the estimated radar, of three beats. The presence of many detected peaks often corresponds to much motion corruption for radar. In this case, however, many peaks are also visible in the belt reference. The example (b), which is part of the samples in the range 7–9 peaks, is characterized by more radar signal corruption with respect to (a). Nevertheless, the robust formulation of $L^{*}$ allows the extraction of peak position and bpm despite the presence of motion corruption. However, for both plots, there are discrepancies in time shift and peak amplitude between radar and belt. As described in Section 3.6 and Section 4.2, the proposed C-VAE topology tries to mediate these challenges but still cannot perfectly reconstruct the belt reference signal. In general, incorrect detection of radar peaks can result in erroneous local predictions of the bpm. In both plots, the corruption flag correctly predicts the motion in correspondence to the false peaks detected. Figure 18 instead shows two edge examples with a relatively low (a) and high (b) belt reference number of peaks per session. In both cases, the model leads to quite different results from the reference ones. Although the bpm estimate does not deviate much from the reference, peaks are detected at incorrect times. For (a), the combination of just a few breathing peaks and motion corruption makes correct prediction challenging. One way to potentially solve this issue would be to collect a lot of edge data and train the model episodically to generalize better in such scenarios. In the example (b), the user breathed much more frequently than in the other sessions, including the training one. This leads to the model’s inability, given the prior acquired knowledge, to generalize to the user scenarios with only one training shot. The time shift between radar peaks is also not seen as corruption by the specific flag, as it is probably not caused by motion.

Table 3 lists the adaptation time for a new user, varying the number of shots in milliseconds using $L^{*}$. The procedure of estimating the adaptation time to new users is explained in Section 4.4. Given four epochs of learning per single user, the algorithm requires a gradually increasing adaptation time as the number of shots increases. Indeed, compared to a 1–shot, the adaptation time is roughly 3 and 7 times longer for 5– and 10–shots. Since the mean $L^{*}$ does not decrease much for 5– or 10–shots (less than 1%), the 1–shot model can be considered the best trade-off. On the other hand, for the 1–shot experiments, there is a bigger variation in the confidence value (up to 5 %). For this reason, we decided to show the 1–shot outcomes in the single-experiment analysis.

The single inference time for MAML $\;2^{nd}$ experiments, as discussed in Section 4.4, is independent by the number of shots used for user adaptation. Accordingly, we calculated an average value over all repetitions of the 1–, 5– and 10–shots experiments, already averaged over the last 300 test evaluations of each. The computed value of single inference time for MAML $\;2^{nd}$ is 4.30 ms.

### 5.2. Ablation Study

To show the real benefits of robustness to motion corruption, it is also important to compare it with training that does not take this information into account when figuring out the loss function. This can be conducted by comparing the results obtained with $L^{*}$, with the loss *L* presented in Section 4. The average values over three experiments for loss comparison are given in Table 4. As can be seen from the results, the mean $L^{*}$ turns out to be less than half despite the fact that the two formulated losses are characterized by the same magnitude for the three components to minimize. Moreover, the 95 % confidence shows that the loss *L* does not become more accurate as the number of shots increases. Probably, without limiting the learning for corrupted training examples, as for $L^{*}$, the model also learns information that is not useful for respiration estimation.. Aside from the values, restricting feature extraction to pure breathing information improves learning and model prediction by incorporating the amount of motion corruption in loss formulation.

Another important feature to analyze is how the performance of the chosen C-VAE topology varies as a function of the model size. This can be accomplished by varying the size of the latent space, which represents the size of the extracted features. MAML $\;2^{nd}$ 1–shot experiments were carried out with latent space values ranging from 16 to 128. Table 5 shows the values of $L^{*}$ and the number of trainable parameters as a function of the latent dimension. The mean $L^{*}$ reaches the minimum for a latent dimension of 32, which was also selected for all experiments in Section 5.1. A latent dimension of 16 seems to not be enough to extract all useful breathing features from the radar phase. A dimension of 64 brings similar mean values to 32 at the expense of twice as many parameters. Looking also at the values at 95%, both a latent space of 64 and 128 lead to a visible degradation of precision. This means that the features extracted for many of the evaluation episodes, tend to overfit the training data and thus fail to generalize well to test users.

### 5.3. Results on Various Optimization-Based Algorithms

Using the same C-VAE topology, the performance of MAML $\;2^{nd}$ Order can be compared to that of other cutting-edge Meta-L algorithms.

We propose comparing MAML $\;1^{st}$ (first-order model) [30], Reptile [44], and an improved in-training stability version of MAML based on some contributions from Antoniu et al. [45]. We call MAML^+^, the stabilized version of MAML that incorporates multi-step loss optimization (MSL), derivative-order annealing (DA), and meta-optimizer learning rate cosine annealing (CA). We trained such algorithms following the same episodic training and evaluation setup as defined in Section 5.1, for MAML $\;2^{nd}$. Reptile episodic training was carried out with a batch size of 2 and an inner learning rate of 3 × 10^−5^. The outer step weight update has been conducted with a meta step size of 0.4. For MAML^+^, a value of 1.7 × 10^−5^ is chosen as the initial value for the outer step learning rate before cosine annealing.

The average accuracy values for test users over three repetitions of the experiment are given in Table 6. For 1– and 5–shots, MAML algorithms perform better than Reptile. MAML 2^nd^ produces the lowest average value of $L^{*}$ for a single shot, allowing for better reconstruction of respiration signals. For 5–shots, the MAML^+^ algorithm guarantees the best average result. The latter, however, lacks precision, leading to a broad 95% confidence interval in the 10–shot approach and even decreasing the learning rate in the inner step. Most likely, second-order learning, coupled with training that tends to be more selective as episodes progress, leads the model to give more weight to motion corruption features. This leads to instability when multiple training samples are employed and thus decreases performance. MAML $\;2^{nd}$ and MAML $\;1^{st}$ are tied as the algorithms with the lowest mean $L^{*}$ for 10 shots. For all algorithms except Reptile, it can be seen that there is no marked decrease in the mean $L^{*}$ as the number of shots increases. By having more data available, this behavior could be countered by using only the least corrupted data for training.

The adaptation procedure adopted for the other algorithms is the same as that of MAML $\;2^{nd}$, illustrated in Section 4.4. As the algorithms vary, only the procedure for computing the generalization parameters in the Meta-L stage changes. The parameters in the evaluation phase are algorithm specific, but the adaptation always employs first-order gradient optimization. This means that the adaptation time is independent of the chosen algorithm, since what changes are only the values of the parameters. Thus, there is no real difference in adaptation time between the chosen algorithms with respect to the value provided in Section 5.1. The same is true for the single-sample inference time.

## 6. Conclusions

In this paper, we present a user-adaptable and non-contact solution for respiration signal estimation using a 60 GHz FMCW radar. This system is mainly intended for office work-desk applications in a distance range of 20 to 40 cm, characterized by little user motion. This solution, while not as accurate as user-contact estimation approaches, shows how radar can potentially be employed to non-contact monitor respiration rates. The episodic learning approach eases the system’s adaptation to new users through short model adaptation sessions. The estimated respiratory rate may be used for anomaly detection related to the specific user to whom the system is tailored. Thanks to a variational autoencoder, the topology employed can extract respiration features from the radar phase signal, using as a reference for reconstruction, the signal collected with a respiration belt. Although the belt could be used on its own for respiration estimation, it would not allow non-contact estimation. Through this approach, the belt can serve only in user-specific learning to enhance radar predictions. The cost function of the model is suitably modified by constraining feature generation to the respiration information, to avoid learning motion corruption information. In addition, a direct estimation of corruption in the collected data sessions allows for improved learning and model estimation in breath signal generation. The whole system presented represents the first step toward a possible non-contact solution for estimating multiple vital parameters that is adaptable quickly and has cutting-edge performance for new users. The radar solution by sensing millimeter displacements could also be used for estimating cardiac signals or the presence of muscle tremors caused by potential diseases.

Although this solution offers several innovative advantages, it also has the disadvantage of relying, only during adaptation, on a breathing belt used as a reference. We placed the constraint so that such a reference sensor depends little on degradation caused by user motion. The generated models also do not perform particularly well for users with respiratory rates significantly higher or lower than the standard 7 to 9 beats per 30 s. This is mainly due to only a few reference examples available in meta-learning, not enough for proper generalization. Radar information is also easily corrupted by long movements in the recording sessions Further, the use of multiple training examples for user adaptation results in improvements for out-of-standard respiratory rates but can degrade performance within the standard range itself. Therefore, substantially motion-corrupted sessions should still be discarded and not used for adaptation. Sensor fusion systems and discarding corrupt sessions could improve performance under these circumstances.

Future work will focus on benchmarking the presented approach against other non-contact solutions, comparing the Meta-Learning solution with transfer learning and adapting the system to other environments, such as outdoors. An additional important aspect that will be analyzed is the variation in model performance per user and bpm as the level of motion corruption in the sessions changes. Another intriguing possibility would be to test and improve the solution on users with respiratory dysfunction in order to assess its benefits and drawbacks.

## Figures and Tables

**Figure 1 sensors-23-00804-f001:**
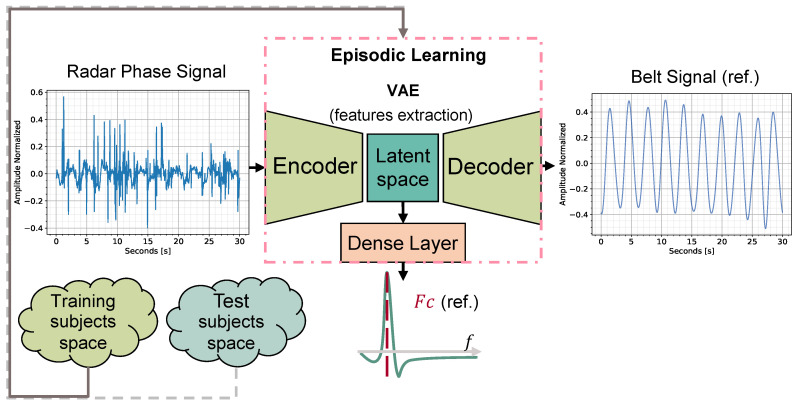
For each learning episode, a training subject is randomly sampled. For each training shot, the radar phase information is mapped to the reference belt signal (ref.) via a C-VAE. Through a dense layer, the ANN also tries to regress the extracted respiration Fc, learning from the ideal belt Fc. The latent space mapping is thus constrained to the Fc, whose estimate is also used in the prediction phase.

**Figure 2 sensors-23-00804-f002:**
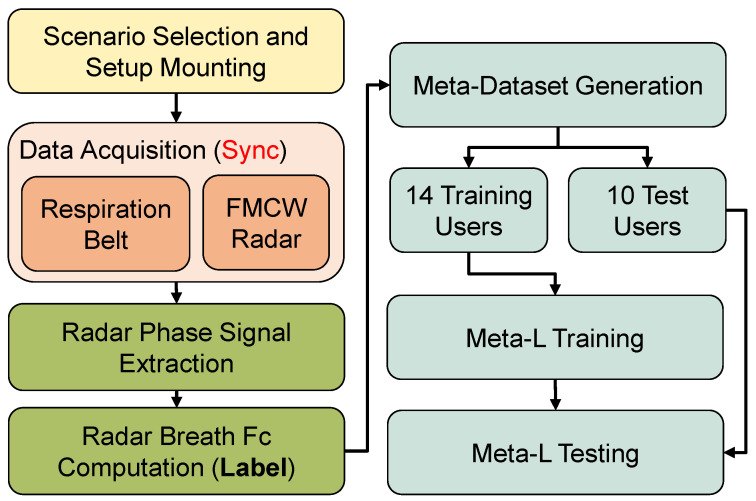
The diagram shows the main steps of the implementation. For a chosen scenario (room and user), several data sessions with synchronized radar and a reference respiration belt are collected. For multi-output ANN, the labels consist of belt reference signals and the central breath frequencies, estimated from the pure belt reference. The data from fourteen users are then used to train an ANN episodically using Meta-L, while the data from the remaining ten users are solely used for testing.

**Figure 3 sensors-23-00804-f003:**
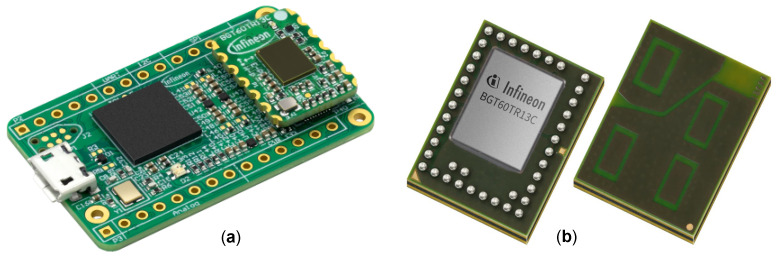
The *BGT60TR13* radar system (**a**) delivers filtered, mixed, and digitized information from each Rx channel. The *BGT60TR13C* radar (**b**) is mounted on top of the evaluation board.

**Figure 4 sensors-23-00804-f004:**
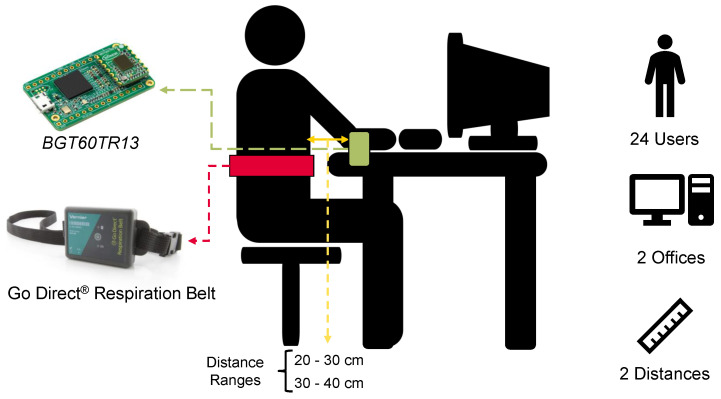
Recording Setup. A synchronized radar system and respiration belt are used to collect 10 30-second sessions per user and distance. The distance ranges used in data collection (up to 30 or 40 cm), refer to the distance between the chest and the radar board.

**Figure 5 sensors-23-00804-f005:**
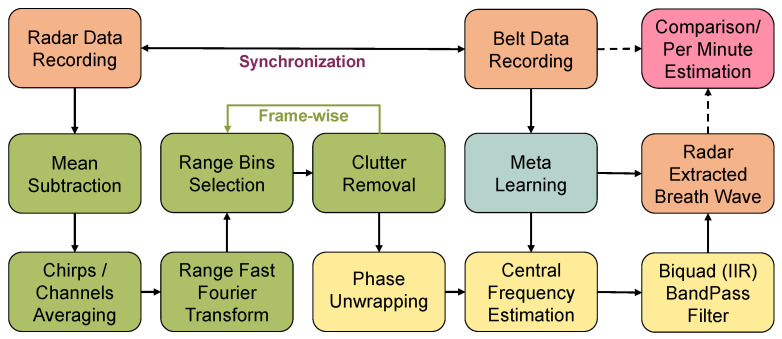
Preprocessing pipeline. First, the phase information is unwrapped from the raw radar data. The respiration signal and Fc are then estimated by Meta-L, exploiting only in the training phase the data collected with the respiration belt.

**Figure 6 sensors-23-00804-f006:**
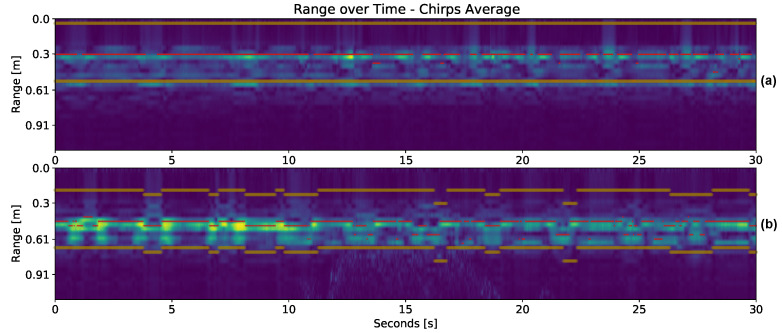
Lines in yellow indicate the defined range bin limits and, in red, the detected maximum bin per frame. Range plotting is generated after clutter removal. In (**a**), the subject did not move much during the session. In (**b**), the range limits vary according to the user’s distance from the radar board.

**Figure 7 sensors-23-00804-f007:**
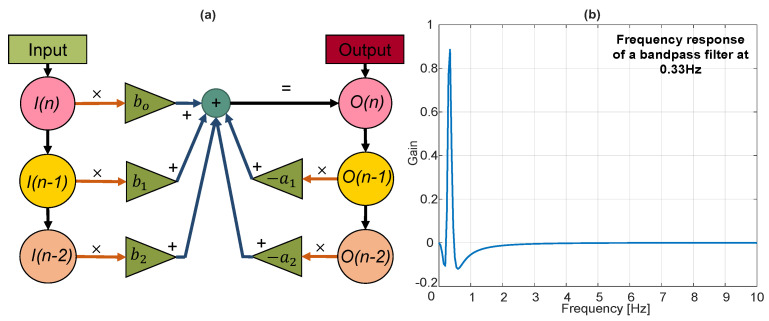
Band-pass bi-quadratic filter. The diagram (**a**) depicts the linear flow of the biquad filter, where the output O(n) at time instant *n* is determined by the two previous input *I* and output *O* values. Instead, a gain vs. frequency plot of a biquad band-pass filter obtained for a *Q* of $\sqrt{2}$ and fs of 20, over an Fc of 0.33 Hz, is shown as a reference in (**b**).

**Figure 8 sensors-23-00804-f008:**
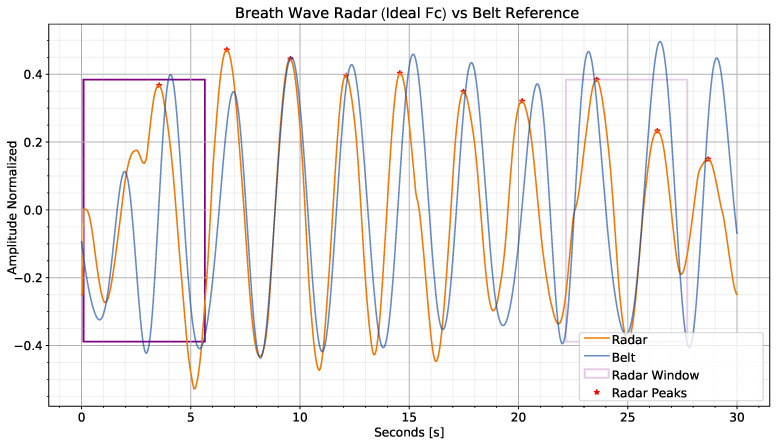
Example of sliding window generation for instant bpm estimation on a recorded session The radar signal has been filtered using the ideal belt, Fc. The radar, as opposed to the belt, is not connected to the user during recordings, but to the desk. This results in the local shift of signal breathing peaks due to the millimeter movements of the user. The window (in purple in the plot) is shown paler on the two peaks closest to the calculated peaks’ mean distance. It is also possible to notice some slight corruption at the beginning of the session due to user motion.

**Figure 9 sensors-23-00804-f009:**
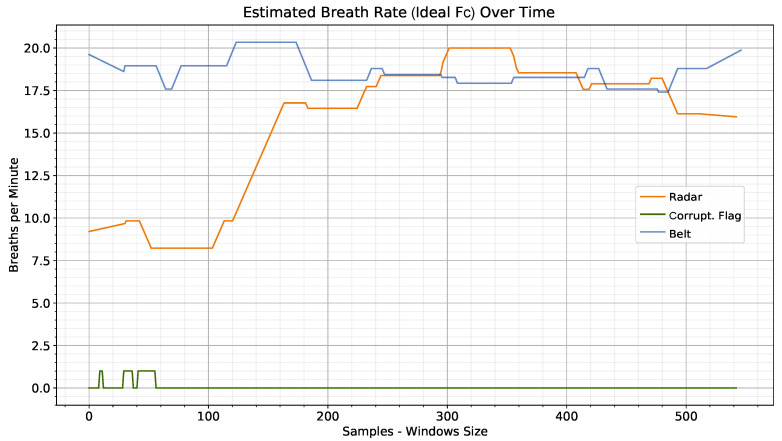
Comparison of instantaneous bpm between respiration belt and radar (with ideal Fc) for a recording session. The x-axis corresponds to the difference between the number of frames in the session and the sliding window length. The radar signal corruption flag variable is plotted in green. At the beginning of the session, the radar signal is motion-corrupted (as shown in Figure 8) and thus does not lead to a reliable bpm. On the other hand, for the workplace use case, the reference belt signal is more robust to motion. In this case, the motion performed was the movement of the hands toward the desk.

**Figure 10 sensors-23-00804-f010:**
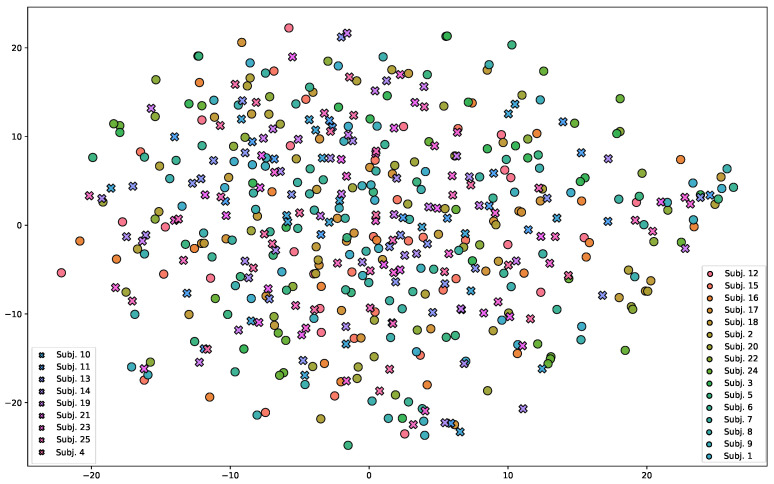
Two-component t-SNE representation of the *Breath Meta-Dataset* radar data. The circles represent the training users, while the crosses represent the testing users for the Meta-L. No user-specific feature clusters are visible under the t-SNE assumptions. The t-SNE was obtained with a perplexity of 20 and 7000 iterations [42].

**Figure 11 sensors-23-00804-f011:**
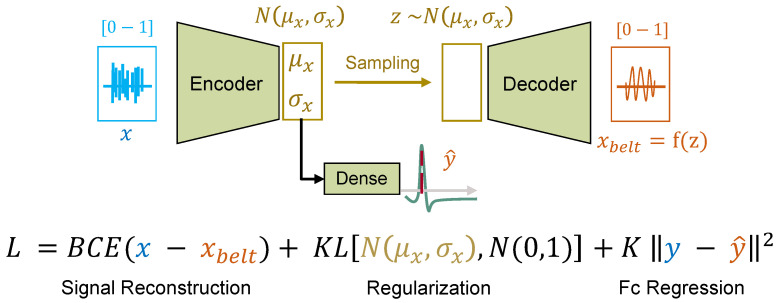
Graphical representation of single-episode learning with C-VAE. The unwrapped radar phase is mapped to the respiration belt signal using the signal reconstruction term. The regularization term makes the latent space closer to a standard multivariate normal distribution. Fc regression allows the parameterization to depend on the respiration signal.

**Figure 12 sensors-23-00804-f012:**
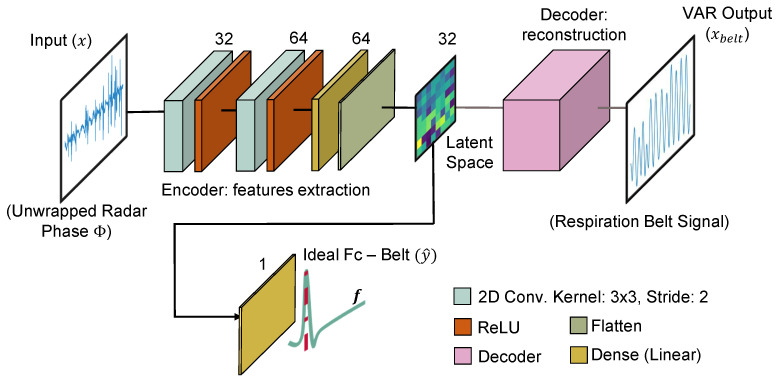
Chosen C-VAE topology. The latent space representation is constrained by both the reconstruction of *x* with respect to the $x_{belt}$ reference and the ideal Fc of breathing *y*. The decoder layers are an up-sampled mirror version of the encoder layers.

**Figure 13 sensors-23-00804-f013:**
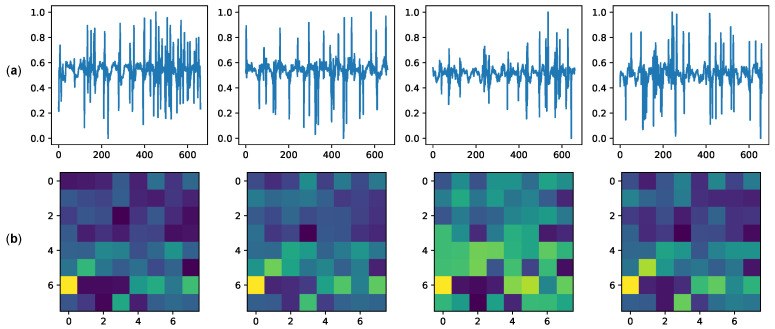
Examples of latent space generation. Examples of radar phase input (**a**) and generated latent spaces (**b**), size 32, are shown. The latent spaces are obtained after the model generalization training. Each 8x8 representation consists of the mean values μ and the standard deviations σ. Starting from the top of the representations toward the right, the first 32 pixels represent μ values, while the last 32 are those of σ.

**Figure 14 sensors-23-00804-f014:**
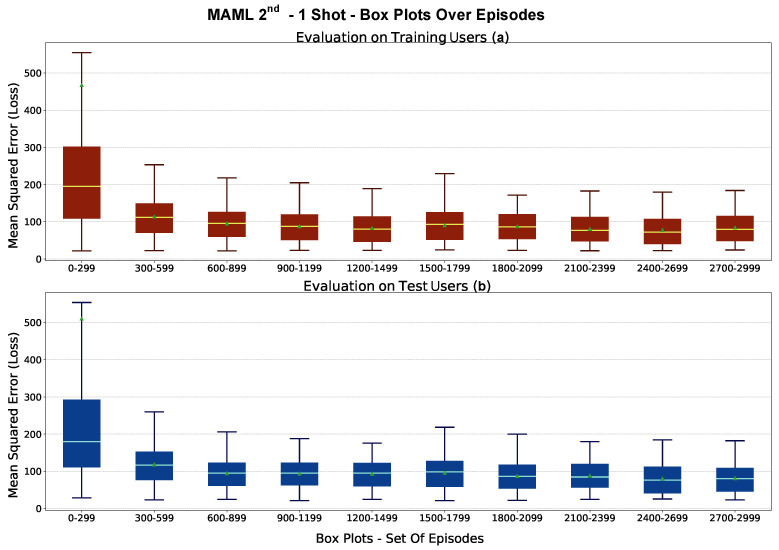
MAML $\;2^{nd}$ 1–shot experiment, Box Plots. Learning trends of Meta-L, box plots versus episodes (evaluation loop) for the *Breath Meta-Dataset*. The box in (**a**) depicts the trend for users in the training set ($T_{r}$ tasks). In (**b**), the trend for the users of the test set ($T_{v}$ tasks)is shown. The box’s mid-line represents the median value, while the little green triangle represents the mean.

**Figure 15 sensors-23-00804-f015:**
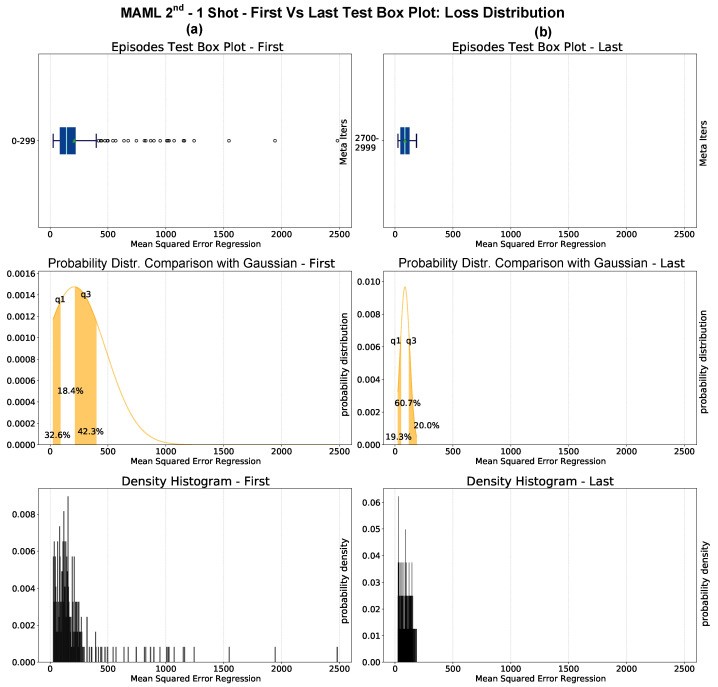
MAML $\;2^{nd}$ 1–shot experiment histograms for the first (**a**) and last (**b**) set of 300 episodes. The box plots in the topmost plots also contain outliers as small circles outside the whiskers. The mid-plots show an approximation to the Gaussian distribution. The lower plots show the true histograms, which do not underlie a Gaussian distribution. The q1 and q3 represent the first and third quartiles, respectively.

**Figure 16 sensors-23-00804-f016:**
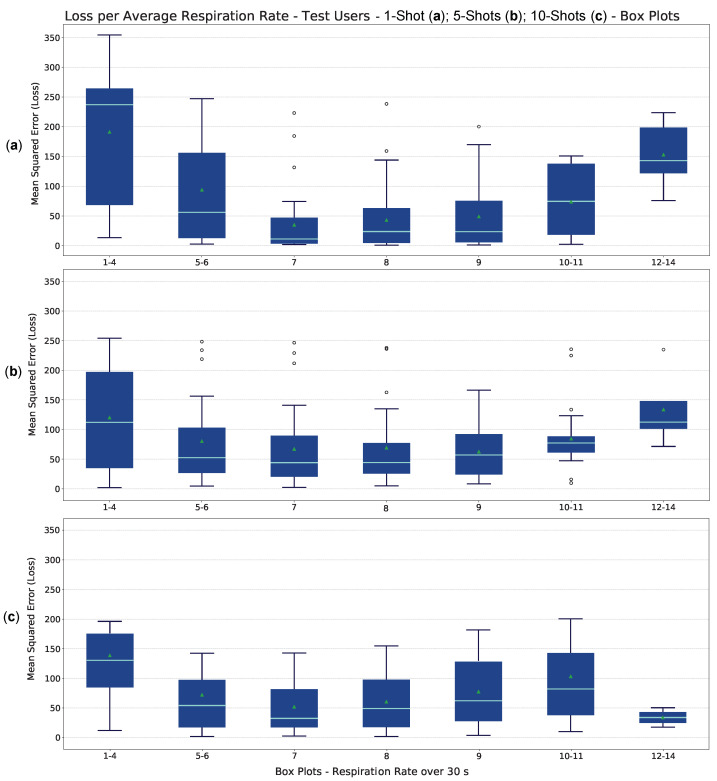
Loss ($L^{*}$ ) as a function of the number of detected breathing spikes over the 30 s sessions for the 10 test users. The base of the box plots with non-uniform ranges was chosen so as to have at least 4 examples for the least common classes (1–4 and 12–14). The upper plot is obtained by fitting the 1–shot Meta-L model (**a**) to new users, while the middle and lower plots are obtained by 5– (**b**) and 10– (**c**) shots adaptation, respectively. For the first two plots, the circles that lie outside the box plots whiskers represent the outliers. Plot (**c**) shows no visible outliers.

**Figure 17 sensors-23-00804-f017:**
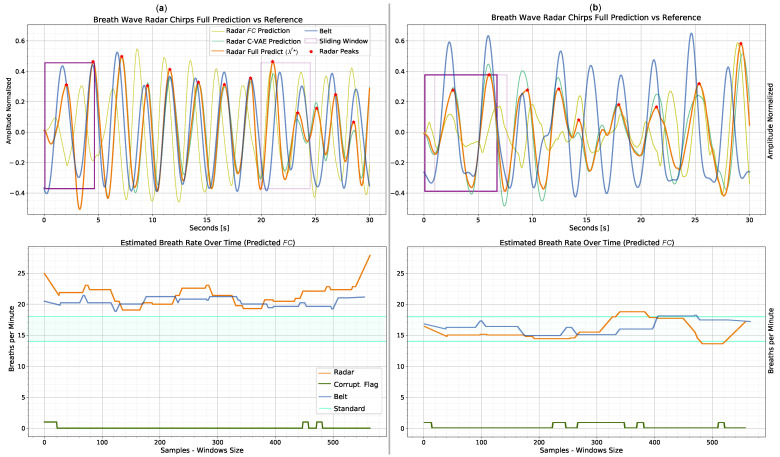
Standard prediction examples obtained post 1–shot test user-adaptation with MAML $\;2^{nd}$. The top plots show the prediction ${\hat{x}}^{*}$ versus the respiration belt reference, while the bottom plots display the estimated bpm and corruption flag. Legends, which also apply to the plots on the right, are placed in the plots on the left. An example of optimal prediction with radar information characterized by little motion corruption is shown in (**a**). The respiration signal is recovered even in the presence of some corruption, as in (**b**), thanks to the $L^{*}$ formulation.

**Figure 18 sensors-23-00804-f018:**
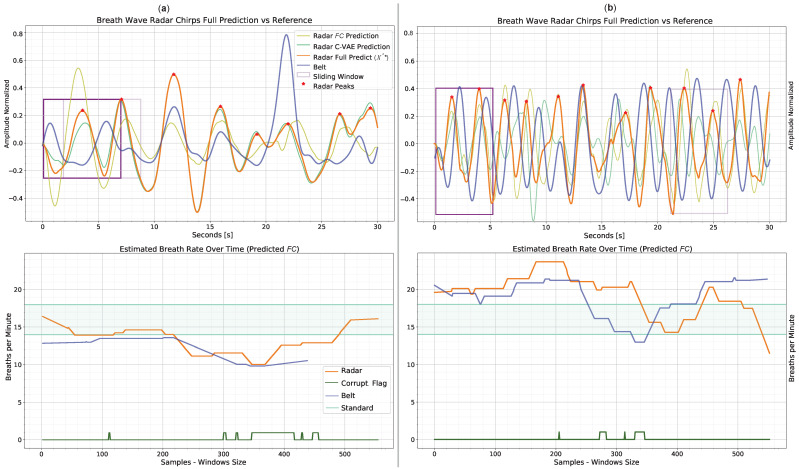
Edge prediction examples obtained post 1–shot test user-adaptation with MAML $\;2^{nd}$. The top plots show the prediction ${\hat{x}}^{*}$ versus the respiration belt reference, while the bottom plots display the estimated bpm and corruption flag. Legends, which also apply to the plots on the right, are placed in the plots on the left. In (**a**), there are six visible peaks in the belt signal (blue), while in (**b**) there are thirteen peaks. In these examples, the algorithm performs less well than in standard cases. This is mainly due to the lack of edge data as prior knowledge during episodic learning. In the bpm estimation in the example (**a**), a shorter estimate can be seen for the belt than for radar. This is due to the computation of two distinct windows between radar and belt, as explained in Section 3.6.

**Table 1 sensors-23-00804-t001:** *BGT60TR13C* radar board, parameters configuration for breath sensing.

Symbol	Quantity	Value
$N_{Tx}$	number of transmitters	1
$N_{Rx}$	number of receivers	3
$N_{c}$	number of chirps	2
$n_{s}$	samples per chirp	200
$f_{0}$	center freq.	60 GHz
$F_{s}$	sampling freq. ADC	2 MHz
fps	frames per second	20 Hz
$t_{c}$	chirp time duration	150 µs
$B_{w}$	bandwidth	[58, 62] → 4 GHz

**Table 2 sensors-23-00804-t002:** MAML $\;2^{nd}$ experiments, average $L^{*}$ over the last 300 episodes of test tasks $T_{v}$ evaluation, averaged over 3 repetitions with 95% confidence intervals.

Loss / N–Shots	1–Shot	5–Shots	10–Shots
$L^{*}$	84.11 ± 6	83.92 ± 1	83.39 ± 1

**Table 3 sensors-23-00804-t003:** MAML $\;2^{nd}$ experiments, average adaptation time over the last 300 episodes of test tasks $T_{v}$ evaluation, averaged over 3 repetitions using $L^{*}$, in milliseconds.

Time N–Shots	1–Shot	5–Shots	10–Shots
Adaptation Time [ms]	797	2,614	5,877

**Table 4 sensors-23-00804-t004:** MAML $\;2^{nd}$ experiments, average *L* and $L^{*}$ over the last 300 episodes of test tasks $T_{v}$ evaluation, averaged over 3 repetitions, with 95% confidence intervals.

Loss / N–Shots	1–Shot	5–Shots	10–Shots
L (No Corrupt.)	226.30 ± 5	224.53 ± 5	221.97 ± 5
$L^{*}$ (Corrupt.)	84.11 ± 6	83.92 ± 1	83.39 ± 1

**Table 5 sensors-23-00804-t005:** MAML $\;2^{nd}$ 1–shot experiments, average $L^{*}$ and trainable parameters with varying latent dimension. The $L^{*}$ values are obtained over the last 300 episodes of test tasks $T_{v}$ evaluation. The results are provided with 95% confidence intervals, averaged over 3 repetitions.

Parameters / Latent Dim.	16	32	64	128
$L^{*}$	86.75 ± 5	84.11 ± 6	84.31 ± 14	85.19 ± 34
Trainable Params.	382,658	739,074	1,451,906	2,877,570

**Table 6 sensors-23-00804-t006:** Optimization-based experiments comparison, average $L^{*}$ over the last 300 episodes of test tasks $T_{v}$ evaluation, averaged over 3 repetitions with 95% confidence intervals.

Algorithm N–Shots	1–Shot	5–Shots	10–Shots
Reptile	100.02 ± 2	90.78 ± 2	86.95 ± 1
MAML $\;1^{st}$	86.52 ± 5	83.68 ± 1	83.45 ± 1
MAML ${\;}^{+}$	85.86 ± 10.7	82.9 ± 3	88.16 ± 15
MAML $\;2^{nd}$	84.11 ± 6	83.92 ± 1	83.39 ± 1

## Data Availability

The data are not publicly available due to internal company board policy.

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
