# Peer review of "Few-Shot User-Adaptable Radar-Based Breath Signal Sensing"

_sensors, 2023, doi:10.3390/s23020804_

Round 1
Reviewer 1 Report
1. In many places throughout the manuscript there are unnecessary capital letters; some examples are: Heart Rate, Respiration Rate, Telehealth, Frequency Modulated Continuous Wave, Field of View Artificial Intelligence, Machine Learning and many more. This should be corrected.
2. Fc is a symbol such as Fs (used in Table 1) and should be denoted with the same font as the rest of the symbols in the manuscript.
3. Line 269: What do SIF(n) and ninNrx denote?
4. The intervals should be denoted with [a,b] and not with [a – b].
5. Equation 4: The symbol SR(s) bar is used on both sides of the equation what can be misunderstood.
6. Line 293: What is “peak-to-clutter information”? How is it determined?
7. In many places the fonts used to write the same symbol are different; e.g. see symbols n, x and y in Equation 5 and the description below that equation. Please correct that.
8. Line 297: what are the values of a0, a1, a2, b0, b1, and b2?
9. In Section 3.5.1 y is used to denote filtered signal x while in Section 4.1 it is used to denote the central frequency extracted from the belt. Please make the system of the symbols consistent.
10. In Section 4.1 x hat is said to denote reference data of the respiration belt while in Section 4.3 it is said to denote estimated output. Please make the system of the symbols consistent.
11. Figure 13: How exactly are the 8x8 representations constructed from the mean and standard deviation values?
12. Figure 17 is unreadable because of the colours used: first two lines are not visible enough.
13. Line 521: “–5” is not the power of e; the correct notation is 3e–5.
14. Line 522: please use minus sign in 17e–4.
Author Response
Dear Reviewer,
Thank you for taking the time to review and emphasize the value of our work. We really appreciate it.
As an attachment to this note, the response to the Review Report. The document consists of two parts. First, a point-to-point response to all the comments. Second, the updated manuscript, with all relevant changes highlighted in blue.
Best Regards,
Gianfranco Mauro et al.

Reviewer 2 Report
Please, see the attached file.

Author Response

(The authors gave the same response as above.)

Round 2
Reviewer 2 Report
The authors responded point by point to all comments. They revised several sections of the paper with details and explanations, thus improving the comprehension and reproducibility of the study. Some of the figures were also changed to increase readability. The changes have increased the paper's overall quality that is therefore acceptable in its present form.